# Experimental Models to Study the Pathogenesis and Treatment of Mucormycosis

**DOI:** 10.3390/jof10010085

**Published:** 2024-01-22

**Authors:** Ronen Ben-Ami

**Affiliations:** Department of Infectious Diseases, Tel Aviv Sourasky Medical Center, Faculty of Medicine, Tel Aviv University, Tel Aviv 64239, Israel; ronenba@tlvmc.gov.il; Tel.: +972-3-6974347

**Keywords:** mucormycosis, models, animal, virulence, diabetic ketoacidosis

## Abstract

Mucormycosis presents a formidable challenge to clinicians and researchers. Animal models are an essential part of the effort to decipher the pathogenesis of mucormycosis and to develop novel pharmacotherapeutics against it. Diverse model systems have been established, using a range of animal hosts, immune and metabolic perturbations, and infection routes. An understanding of the characteristics, strengths, and drawbacks of these models is needed to optimize their use for specific research aims.

## 1. Introduction

Mucormycosis is a life-threatening invasive fungal disease caused by members of the order Mucorales and characterized by angioinvasion, tissue necrosis, and relentless extension across tissue plains [1,2]. *Rhizopus*, *Mucor*, and *Lichtheimia* species account for >90% of clinical cases [2]. Populations at risk include patients with compromised immunity due to cytotoxic therapy-associated neutropenia or corticosteroid treatment, especially in the context of hematological malignancies, stem cell or solid organ transplantation, patients with poorly controlled diabetes mellitus and ketoacidosis, and survivors of traumatic injury [3,4,5,6]. Recently, the combination of diabetes mellitus, COVID-19, and treatment with corticosteroids has been implicated in a large-scale outbreak of mucormycosis, predominantly in India [7,8]. Reported mortality rates are high, with only ~40% of patients surviving 12 weeks from diagnosis [3,6,9]. The incidence of mucormycosis may be increasing, especially in patients with hematological malignancies and recipients of stem cell transplantation [9,10,11,12,13]. In these populations, mucormycosis is driven by increasingly complex and profound immune suppression, as well as a shift to breakthrough fungal disease in patients receiving mold-active antifungal prophylaxis [14]. Preclinical and clinical assessment of drugs and novel drug candidates is of paramount importance [15]. However, conducting controlled clinical trials of mucormycosis is challenging, given the overall rarity of the disease and patient heterogeneity in terms of infecting Mucorales species, routes of infection, predisposing comorbidities, pharmacotherapy, and surgical interventions. To date, there have been no adequately powered randomized controlled clinical trials for mucormycosis, and most therapeutic interventions are based on observational data and expert opinion [16]. Animal models offer opportunities to extend in vitro observations and establish in vivo drug activity, before proceeding to controlled clinical trials [17,18,19,20,21,22,23,24,25,26,27]. Moreover, such models have been instrumental in elucidating key aspects of the pathogenesis of mucormycosis [19,28,29,30,31,32,33,34,35,36].

In selecting an animal model system, researchers must examine relevant attributes that match their research questions (Table 1). Important aspects to consider include: 1. throughput; 2. route of infection mimicking clinical human disease; 3. host proximity to normal and impaired human immunity against Mucorales; 4. options for drug delivery and pharmacokinetic analyses; 5. ability to monitor infection kinetics; 6. ability to perform single or repeated blood and tissue sampling; 7. amenity to genetic manipulation; 8. cost; and 9. ethical considerations. In this review, the features, applications, and limitations of various animal models of mucormycosis will be discussed.

## 2. Mammalian Models of Mucormycosis

Mammalian hosts provide the closest anatomical and physiological models for human mucormycosis. Mice are by far the most frequently used experimental animals, due to their relatively low cost and ease of handling. Larger rodents, including rabbits, guinea pigs, and rats have also been utilized [37,38]. Mammalian models can be usefully categorized according to the immune–metabolic perturbation used to enhance susceptibility to Mucorales and the route of infection.

### 2.1. Modeling Immune-Metabolic Perturbations Predisposing for Mucormycosis

Although mucormycosis can occur in individuals with apparently healthy immune systems, most cases reported in large case series fall within two risk groups [3,5,6,39]: 1. patients with uncontrolled diabetes mellitus, specifically those with diabetic ketoacidosis, and 2. patients with hematological malignancies, including stem cell transplantation recipients. Within the second group, neutropenia and high-dose corticosteroid treatment are the main predisposing factors [40,41], which can be modeled separately. Other risk groups include recipients of solid organ transplantation and other patients with inflammatory disorders treated with immunomodulatory agents.

#### 2.1.1. Neutropenia

Experimental data indicate that phagocytes, including neutrophils and macrophages, are critical for host defense against Mucorales [40,42,43,44]. In the immune-competent host, alveolar macrophages ingest inhaled sporangiospores and prevent their germination. Neutrophils damage and kill Mucorales by generating oxidative metabolites and cationic peptides [42]. Depletion or impaired function of innate phagocytic cells leads to increased susceptibility to mucormycosis. In patients with hematological malignancies, neutropenia predisposes to mucormycosis, and recovery from neutropenia is an important prognostic factor [40].

Neutropenia and susceptibility to mucormycosis can be induced in rodents using cytotoxic drugs, such as the alkylating agent cyclophosphamide (for mice) and the antimetabolite cytarabine (for rabbits), or more specifically by using monoclonal neutrophil depleting antibodies [45,46,47]. Cyclophosphamide at a cumulative intraperitoneal dose of 250 mg/kg (150 mg/kg on day 1 and 100 mg/kg on day 4) induces profound neutropenia (≤10 neutrophils/mm^3^) on day 4, which persists on days 5 and 6 [46,47]. Repeated doses may be necessary to maintain neutropenia over longer periods [47]. Available monoclonal antibodies used to deplete neutrophils target antigens Ly6G (anti-Ly6G clone 1A8) and both Ly6G and Ly6C (anti-Gr1 clone RB6-8C5) [45]. Ly6G is a specific neutrophil marker, whereas Ly6C is additionally expressed on monocytes, macrophages, T cell subsets, eosinophils, and endothelial cells; thus, anti-Ly6G antibodies induce specific neutrophil depletion [45]. RB6 antibodies, at doses of 7.5 µg or higher, induce absolute neutropenia after 1 day, with rapid recovery 1 day later [48]. Interestingly, while cyclophosphamide alone renders mice susceptible to inhalational Mucorales infection, selective depletion of neutrophils using monoclonal antibodies does not [49], suggesting that concurrent inhibition of alveolar macrophage function is essential.

#### 2.1.2. Corticosteroid Treatment

Mucormycosis has emerged as a major complication of solid organ and stem cell transplantation, typically occurring late (>3 m) after transplantation [41]. High-dose corticosteroids, usually administered for graft-versus-host disease, were identified as an independent risk factor for mucormycosis in stem cell transplant recipients, but not in solid organ recipients [41]. Corticosteroid treatment, in combination with uncontrolled diabetes mellitus, was present in a large proportion of patients with COVID-19-associated mucormycosis [8]. For patients with mucormycosis, corticosteroid use is associated with a 4-fold higher risk of death, compared to untreated controls [50].

Immunosuppression with cortisone acetate alone is sufficient to produce pulmonary and disseminated mucormycosis after intranasal inoculation of sporangiospores in mice [17,51,52,53]. Various corticosteroid regimens have been described. For example, subcutaneous cortisone acetate, 250 mg/kg every 3 days, starting 4 days before infection until day 5 after infection [52], or intraperitoneal hydrocortisone acetate, 250 mg/kg every 2 days, starting 4 days before infection until day 4 after infection [51]. In addition, corticosteroids are often added to cytotoxic regimens to enhance suppression of resident tissue macrophages (e.g., cortisone acetate, 250–300 mg/kg by subcutaneous injection, 1 day before infection) [18,22].

#### 2.1.3. Diabetes Mellitus and Ketoacidosis

Uncontrolled hyperglycemia and ketoacidosis are important risk factors for mucormycosis, frequently manifesting as rhino-orbital-cerebral infection [6]. During ketoacidosis, iron disengages from transferrin, and the released free iron impairs the function of neutrophils and undermines their ability to damage and kill Mucorales hyphae [54,55,56].

Protocols for pharmacologically induced insulin-dependent diabetes mellitus and ketoacidosis in mice have been extensively described and validated [29,31,32,57,58,59]. Following a fasting state of 4 to 8 h, mice receive a single injection of either streptozotocin (190–250 mg/kg in citrate buffer intraperitoneally) or alloxan (100 mg/kg intravenously). After injection, animals receive drinking water with 10% sucrose for 24 h. The development of hyperglycemia and ketoacidosis throughout the experiment is monitored by measuring glucose and pH in blood and ketones in urine [29,57]. Ketoacidosis typically develops 7 days after streptozotocin injection, and inoculation of Mucorales sporangiospores is performed 10 days after the injection.

Mucorales species-dependent differences in virulence were observed in mice with ketoacidosis. Specifically, *Rhizopus* species are able to establish infection with no additional immunosuppression, whereas the addition of cortisone acetate is required for infection with *Lichtheimia* species [32,57]. These observations are congruent with the clinical association of *Rhizopus* species with rhino-orbital-cerebral mucormycosis in patients with diabetes mellitus [57].

### 2.2. Experimental Models of Mucormycosis Clinical Syndromes

#### 2.2.1. Sinopulmonary Mucormycosis

Most human cases of mucormycosis result from the inhalation of airborne spores, which deposit in the upper or lower airways, causing invasive sinusitis (rhino-orbital-cerebral mucormycosis) and pulmonary mucormycosis. Sinopulmonary mucormycosis can be recapitulated in mice pre-treated with cytolytic agents, cortisone acetate, or streptozotocin [17,18,19,20,21,26,30,31,33,37,38,51,52,53,60,61,62,63,64,65], as outlined above. Either male or female mice can be used for neutropenic or diabetic models, as only minor sex-dependent differences were observed in susceptibility to infection, and no significant differences in response to treatment or host immune response [66]. The introduction of fungal sporangiospores into mouse airways can be achieved using intranasal or intratracheal inoculation of spore suspension. Inoculation using aerosolization systems, as described for *Aspergillus* species [47], has not been successful for Mucorales, presumably due to the larger spore size [37,67].

Intranasal inoculation is performed under anesthesia. This method is less technically demanding than intratracheal inoculation, and results in reproducible and consistent pulmonary infection [17,18,19,20,21,30,51,52,53,60]. Earlier work conducted using *Blastomyces dermatitidis* spores has shown that 33% of the inoculum CFU are distributed in the lungs 1 h after inoculation [68]. Invasion of the brain occurs frequently in intranasally infected mice and rabbits [53,59], indicating that a significant part of the inoculum is deposited in the paranasal sinuses, making this a sinopulmonary rather than a pure pulmonary infection model. Intratracheal inoculation deposits sporangiospores directly into the lower airways, bypassing the upper airways and avoiding the invasive sinusitis observed with intranasal infection [26,31,33,61,62,63,64,65,67]. Unlike the intranasal route, intratracheal inoculation controls the precise quantity of sporangiospores entering the lungs and might produce more consistently lethal pneumonia in mice [67]. Both surgical and non-invasive methods of intratracheal inoculation are available [67].

Mouse inhalational models are the most widely used and best studied animal models of mucormycosis. Thus, some comparison of the clinicopathologic features of these models, as a function of the immunosuppressive regimen used, is worthwhile.

#### 2.2.2. Neutropenic Inhalational Model

Mucorales can be cultured from the blood of mice 24 h after inhalational inoculation, indicating dissemination to various organs [49]. Death occurs over 7–8 days after inoculation, with variable mortality rates depending on the inoculum size and virulence of the Mucorales species used. For example, higher mortality was observed with *Lichtheimia corymbifera* compared to *Rhizopus arrhizus* [49]. Mortality seems to correlate with dissemination rather than fungal burden in the lungs. Specifically, dissemination to the brain, manifesting as convulsions and vestibular dysfunction, occurs more frequently with *L. corymbifera* versus *R. arrhizus* [49]. Using a high inoculum (e.g., 10^7^ CFU), mortality approaching 100% can be expected with most Mucorales strains [49]. Histopathological examination of lung tissue shows extensive angioinvasive hyphae with hemorrhage and alveolar consolidation [17].

#### 2.2.3. Corticosteroid Inhalational Model

Viable fungi can be recovered from the blood 2 days after inoculation and from the brain 8 days after inoculation [53]. Death typically occurs 6 to 10 days after inoculation [53]. Lung histopathology shows extensive destruction of normal pulmonary architecture with hyphae penetrating into vascular and bronchial structures, hemorrhage, necrosis, neutrophil, and mononuclear cell infiltration [53]. Surprisingly, the fungal burden is similarly high in inhalational models of neutropenic and corticosteroid-treated mice [17,49]. These features differ from those of pulmonary aspergillosis in non-neutropenic cortisone acetate-treated mice, where very few fungal hyphae can be observed in lung sections [69]. In both cases, mice develop an exuberant inflammatory response, suggesting that dysregulated immunity is, at least in part, responsible for tissue damage and death.

#### 2.2.4. Diabetic Ketoacidosis Inhalational Model

Symptoms, such as dyspnea, ruffled fur, and lethargy, appear 3–4 days after infection. Mice and rabbits usually die within 12 to 72 h of symptom onset [57,59]. Dissemination of kidneys, brain, and heart can be observed 5 days after infection [57,58,59]. Lungs show macroscopic areas of tissue necrosis. Histopathology demonstrates a high tissue fungal burden with angioinvasion, thrombosis, ischemic necrosis, purulent inflammation, and destruction of normal architecture [57,58].

In addition to the sinopulmonary model described above, a model of diabetes mellitus-associated rhinocerebral mucormycosis was developed, where the paranasal sinuses are inoculated directly. Streptozotocin-pretreated mice were inoculated with 1–5 × 10^6^ *R. oryzae* sporangiospores in 25–50 µL PBS, and injected into the ethmoid sinuses using a 25-gauge needle [58,70,71]. However, a comparison between intra-sinus and intranasal inoculation showed that both routes result in similar fungal loads in the lungs, whereas fungal load in the brain is significantly higher with the intranasal route [71]. Thus, the need for sinus inoculation to reproduce rhinocerebral mucormycosis appears to be unsupported.

#### 2.2.5. Disseminated Hematogenous Mucormycosis

Inoculation by intravenous injection of sporangiospore suspension leads to disseminated infection in rodents, even without prior immune suppression [37,72,73]. Guinea pigs appear to be more susceptible than mice to intravenous challenge, as evidenced by the lower inoculum required to reproducibly cause metastatic tissue infection and death [37,72]. Notwithstanding, the disseminated infection has been successfully produced in mice with pharmacologically induced neutropenia, corticosteroid treatment, and ketoacidosis [24,74,75]. This type of model may be used when it is desirable to assess fungal dissemination to and proliferation within various organs. As noted earlier, secondary hematogenous dissemination occurs in immunosuppressed and ketoacidotic mice infected via the respiratory route [49,53,57,58]. However, in these cases, dissemination is a late event, with variable frequency [49,53,57,58]. In both clinical and experimental sinopulmonary mucormycosis, hematogenous dissemination occurs after local tissue destruction, necrosis, and angioinvasion have progressed sufficiently to allow significant amounts of hyphae to penetrate the bloodstream [42]. In contrast, primary hematogenous dissemination is extremely rare in clinical mucormycosis, with the notable exception of infection in persons who inject drugs [76].

Intravenous infection bypasses respiratory tract innate immunity, a fact that should be taken into consideration when using these models to study the pathogenesis of mucormycosis. On the other hand, intravenous injection allows for the delivery of a precise inoculum into the bloodstream, an important advantage when comparing the virulence of isogenic Mucorales strains. For example, this approach was used to assess the contribution of specific genes to virulence by comparing the lethality of *Mucor circinelloides* strains with targeted gene deletions [75]. Determination of fungal burden, by means of CFU or genomic DNA normalized for tissue mass, was used to determine dissemination and proliferation in specific organs [75].

#### 2.2.6. Cutaneous and Subcutaneous Mucormycosis

Skin and soft-tissue infections represent the third most common mucormycosis syndrome, after rhino-orbital-cerebral and pulmonary disease [3,4,5,6]. Most cases occur in immune-competent individuals following soft tissue trauma with extensive tissue damage [4]. Intradermal or subcutaneous inoculation of non-immunosuppressed rabbits leads to the formation of circumscribed granulomas with a necrotic center containing spores and neutrophils. These lesions heal spontaneously [77,78]. Thus, cutaneous inoculation in the absence of prior immunomodulation does not lead to the relentless progressive tissue damage typically observed in clinical mucormycosis [4]. In contrast, infection of neutropenic rabbits was characterized by initially expanding lesions, which after 3 days became organized granulomas, presumably in tandem with rising neutrophil counts [77]. Induction of ketoacidosis, even as late as 15 days after infection of non-immunosuppressed rabbits, transformed quiescent granulomas into actively expanding lesions with budding spores, hyphal extension beyond the limits of the granuloma and angioinvasion [77]. Hematogenous dissemination has not been documented in cutaneous models.

Although the feasibility of experimental cutaneous mucormycosis was established six decades ago, this model was more recently optimized to monitor the effectiveness of single and dual drug combinations [22]. Previously established for *Aspergillus* species [79], the model utilizes nude BALB/c mice to allow visual determination of cutaneous lesion dimensions. Using non-nude BALB/c mice and removing hair from the thigh inoculation site is a cost-saving alternative [22]. However, these alternatives are non-equivalent, as nude mice are inherently immunodeficient and require lower doses of cytotoxic drugs to reliably establish infection. Prior to infection, mice receive a neutropenia-inducing regimen of cyclophosphamide (100 mg/kg IP, 4 and 1 days before inoculation and 2 days after inoculation), in addition to cortisone acetate (250 mg/kg SC, 1 day before inoculation) to inhibit the clearance of sporangiospores by tissue macrophages. Inoculation is carried out under anesthesia by injecting 200 µL of spore suspension (e.g., 2.5 × 10^8^ sporangiospores) subcutaneously into the lateral thigh region. A necrotic lesion develops at the inoculation site within 48 h and expands over 7 days of follow-up. This is a nonlethal model in BALB/c mice, and signs of systemic morbidity are scarce throughout the infection. In contrast, disseminated infection with a mortality rate of 40% was observed in subcutaneously infected *Card9^−/−^* mice [80]. At the end of the follow-up period (typically 7–10 days), thigh tissue is excised in its entirety, weighed, homogenized, and analyzed for tissue fungal burden using CFU, qPCR, or chitin content [22,80]. Daily measurement of the skin lesion diameter allows infection to be monitored dynamically over time. An alternative model with footpad inoculation has been described, where infection is monitored as the percent of footpad swelling [80].

Considerations favoring the use of cutaneous versus respiratory mucormycosis models may depend on research objectives. Cutaneous models are well suited to testing the virulence of isogenic Mucorales strains, as precisely identical inoculums are injected into the skin, lesion dimensions are easily compared, and fewer mice are required to demonstrate significant differences in virulence as compared to survival end-point models [79]. Bilateral thigh inoculation of the same animal with different isogenic strains has been performed with *Aspergillus* species [79]. The same characteristics also make the cutaneous model ideal for comparing the antifungal activities of different molecules and the in vivo susceptibility of Mucorales strains [22]. On the other hand, respiratory models are appropriate for studying the pathogenicity and drug response of sinopulmonary mucormycosis, specifically in the context of diabetic ketoacidosis, as well as assessing hematogenous dissemination to different organs.

## 3. Invertebrate Models of Mucormycosis

The use of invertebrates, such as nematodes and fruit files, as model hosts for studying pathogenic fungi, has increased over the past three decades [81]. An important tenet of these models is the understanding that innate immunity to fungi is ancient and evolutionarily conserved even among invertebrates and mammals. Crucially, fungal virulence factors that were characterized in mammalian hosts affect the lethality of infection in invertebrates [81]. The availability of forward genetic tools is an important advantage of invertebrate models [81]. In addition, the low cost, simplicity, and suitability to high-throughput experiments make these models particularly appealing when large-scale testing of multiple fungal strains or antifungal compounds is required. Moreover, invertebrate models are more ethically acceptable than mammalian hosts for screening assays requiring large numbers of animals. It should be noted, however, that because only innate immunity is represented in flies and nematodes, these models generally provide screening systems for fungal virulence, with more detailed analyses incorporating the role of adaptive immunity and organ-specific damage requiring confirmatory testing in mammalian hosts.

### 3.1. Drosophila melanogaster

Drosophila flies have been used extensively to study pathogenic fungi [81]. Infection is performed by injecting the dorsal thorax with a thin sterile needle previously dipped in a suspension of Mucorales sporangiospores [22,34]. Toll-deficient flies are used for most fungal pathogens, as wild-type *D. melanogaster* is intrinsically resistant to fungal infection [81,82]. In contrast, Mucorales sporangiospores injected into the fly hemolymph cause an acute infection with high mortality rates in wild-type flies [34]. Nevertheless, mortality is even higher, approaching 100% for Toll-deficient flies [34]. These observations suggest that similar to humans, the susceptibility of *D. melanogaster* to mucormycosis extends across a wide range of immune competence states.

The Drosophila immune system is well-characterized and amenable to genetic modification [82,83,84,85], making this a convenient model to dissect various aspects of Mucorales pathogenicity [34]. Although the Drosophila model is not suitable for precision pharmacokinetic experiments, simple pharmacologic interventions can be performed by adding drugs to fly feed [19,27,34,86]. For example, the iron chelators deferoxamine and deferasirox had opposite effects on fly mortality and tissue fungal burden, similar to the effect of these molecules in mouse models [19,34]. Similarly, a high-fat diet increased mortality and fungal burden, whereas a regular diet and metformin were protective [86].

### 3.2. Galleria mellonella

Unlike *Drosophila melanogaster*, wild-type larvae of the greater wax moth *Galleria mellonella* are susceptible to infection with multiple pathogenic fungi [87,88,89], making them convenient model hosts of invasive fungal diseases and obviating the need for genetic manipulation. Galleria larvae are inexpensive to purchase and maintain, and the infection model is technically undemanding [88]. Sixth instar *Galleria mellonella* larvae weighing 0.3–0.4 g are used [88]. For experiments requiring drug treatment, dosage is determined by measuring the hemolymph volume of larvae of various weights and calculating the mean hemolymph volume/kg weight by linear regression [90]. The sporangiospore inoculum (usually 10^6^ spores), any drugs being tested, and vehicles serving as infection or treatment controls, are injected diluted in 5–20 µL of insect physiological saline into the larval hemocoel via a hind pro-leg [89,90,91]. Infected larvae are incubated in the dark at 30 °C and monitored daily for survival. Virulent Mucorales strains produce a mortality rate of 85–100% within 4–6 days of infection [89,90,92]. The fungal load can be determined by measuring CFU in a diluted larval homogenate [88]. In addition, the *G. mellonella* humoral immune response can be monitored by determining the expression of select genes encoding for antimicrobial peptides and performing proteomic analysis of these peptides in hemolymph [93]

## 4. Models of Mucormycosis in Non-Mammalian Vertebrate Hosts

### 4.1. Zebrafish

*Danio rerio* (zebrafish) is an appealing model host that combines advantageous features of both invertebrate and mammalian hosts [94]. The zebrafish’s innate and adaptive immune system is comparable to that of humans, including similarly specialized B and T cells and immune signaling molecules [94]. Young fish up to 4 weeks old lack adaptive immunity, facilitating selective studies of innate immunity. Forward genetic tools are available and have been successfully utilized to study mycobacterial infections [95]. Uniquely, the transparency of zebrafish allows non-invasive real-time imaging of infection using fluorescent markers [94].

Zebrafish larvae have been used for elegant studies of the innate immune response to *M. circinelloides* sporangiospores [35]. Injection of viable sporangiospores into the hindbrain ventricle of AB wild-type larvae produced a lethal infection with the hyphal invasion of brain structures monitored in real-time [35]. Immune suppression by pretreatment with dexamethasone or macrophage depletion significantly increased the mortality rate. The zebrafish model allowed for the dynamic visualization of the accumulation of neutrophils and macrophages at the site of infection, where phagocytes were observed to cluster around fungal spores [35]. Moreover, a model of inducible epithelial cell loss, simulating chemotherapy-associated mucositis, was used to show that endothelial extrusion is associated with breaks in cell–cell integrity, uncovering extracellular matrix components that serve as points of entry for *Rhizopus* species [96]. Adult non-immune-suppressed zebrafish are also susceptible to lethal mucormycosis by either intraperitoneal or intramuscular inoculation [36,97], indicating that this model could be used to study fungal interactions with both innate and adaptive arms of the immune system.

### 4.2. Embryonated Chicken Eggs

The avian immune response to fungi resembles that of mammals, including an innate arm consisting of macrophages and neutrophil-like heterophil cells, and an adaptive system capable of producing specific antibodies [98,99]. Embryonated bird eggs are readily available from commercial breeders, are inexpensive, and can be maintained and manipulated without requiring specialized facilities or equipment [99].

Eggs are infected on developmental day 10 via the chorioallantoic membrane. The chorioallantoic membrane is a thin, translucent, highly vascularized membrane consisting of two epithelial layers, that serve for nutrient and gas exchange of the developing embryo and is therefore anatomically and functionally similar to the pulmonary alveolar interface [99]. Because the chicken embryo’s immune system matures with age, infection after developmental day 10 results in low and delayed mortality [100]. Following inoculation, eggs are incubated at 37 °C, and survival is assessed daily by candling (transillumination) over a period of 7 days [92,100]. This model was useful in studying strain- and species-dependent differences in virulence in *Lichtheimia corymbifera* and *Rhizopus* species [92,100].

## 5. Unmet Needs

Experimental sinopulmonary infection using intranasal instillation of spore suspension is by far the most used model for mucormycosis; however, the distribution of the spore inoculum to upper and lower airways is variable and poorly defined. As noted, attempts to perform direct sinus inoculation have nevertheless resulted in sinopulmonary infection [58,70,71]. Thus, none of the currently available models recapitulate human rhino-orbital-cerebral mucormycosis. Similarly, involvement of the central nervous system in inhalational models occurs with variable frequency through direct invasion from the paranasal sinuses and hematogenous spread [53,57,58,59]. A robust system for studying central nervous system mucormycosis has yet to be developed and could serve as a valuable platform to test single and dual drug combinations for this difficult-to-treat entity. Lastly, the growing importance of mucormycosis in patients with severe viral respiratory diseases, as exemplified in the recent epidemic of COVID-associated mucormycosis [7,8], calls for novel strategies to prevent or pre-emptively treat such cases. Animal models for COVID-associated mucormycosis, and indeed for respiratory virus-associated invasive aspergillosis, are currently lacking.

## 6. Conclusions

Animal models have been instrumental in deciphering key aspects of the pathogenesis, host response, virulence, and pharmacotherapy of mucormycosis (Table 2). No single model system is best suited for all these applications. Rather, researchers must make selective use of existing models based on research needs, local expertise, and resources. Given the logistical barriers facing the design and execution of clinical trials of mucormycosis, current and future model systems will likely continue to play a key role in the global effort to develop novel diagnostic tools and therapeutic modalities for mucormycosis.

**Table 1 jof-10-00085-t001:** Features of animal models of mucormycosis.

Feature	Mouse/Rabbit/Rat/Guinea Pig			*Galleria mellonella*	*Drosophila melanogaster*	Zebrafish	Embryonated Chicken Egg
	Sinopulmonary	Hematogenous	Cutaneous				
Throughput	Low	Low	Intermediate	High	High	High	High
Accurate recapitulation of clinical mucormycosis	Yes	No	Partial	No	No	No	No
Drug delivery and pharmacokinetics	Yes	Yes	Yes	Limited	Limited	Limited	Limited
Tools for host genetic manipulation	Limited	Limited	Limited	No	Yes	Yes	No
Cost	High	High	High	Low	Intermediate	Intermediate	Low
References	[17,18,19,20,21,26,30,31,33,37,38,51,52,53,60,61,62,63,64,65,66,67]	[24,37,72,73,74,75]	[22,77,78,80]	[88,89,90,92]	[19,22,27,34,86]	[35,36,96,97]	[100]

## Figures and Tables

**Table 2 jof-10-00085-t002:** Contributions to knowledge by mucormycosis model systems.

Field	Mouse/Rabbit/Rat/Guinea Pig			*Galleria mellonella*	*Drosophila melanogaster*	Zebrafish	Embryonated Chicken Egg
	Sinopulmonary	Hematogenous	Cutaneous				
Host response to Mucorales	Role of endothelial receptor GRP78 [30], iron metabolism [33], EGFR signaling [61]		Role of Card9 [80]		Innate susceptibility of wild-type flies, host gene regulation during mucormycosis, role of Drosomycin (antifungal peptide), iron metabolism and effect of corticosteroid treatment [34] and obesity [86]	Early innate response to Mucorales spores, effect of corticosteroid treatment [35], host gene regulation during mucormycosis [36], susceptibility to infection during simulated mucositis [96]	
Fungal virulence	Voriconazole exposure [51,52], role of Mucoricin [31]	Comparative genomics [75]		Comparative strain virulence testing [88], effect of sporangiospore size [89], thermotolerance [92]	Voriconazole exposure [52]	Effect of sporangiospore size [36]	Comparative strain virulence testing [100]
Diagnostics	PCR [62], *Rhizopus*-specific antigen [65]						
Pharmacotherapy							
In vivo efficacy of novel therapeutic agents	Caspofungin [17], colistin [18], deferasirox [19], statins [60], VT-1161 [63], isavuconazole [64], posaconazole [64,67]	Posaconazole [74]		Rapamycin [90]	Deferasirox [19,34], lovastatin [27]		
Combination therapy	Fluconazole and quinolones [21], isavuconazole and micafungin [26]	Iron chelation, polyenes, and echinocandins [24]	Tacrolimus and posaconazole [22]		Tacrolimus and posaconazole [22], voriconazole and lovastatin [27]		
Pharmacodynamics	Amphotericin B lipid formulations [20]						

## Data Availability

No new data were created or analyzed in this study. Data sharing is not applicable to this article.

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
