# Peer review of "Experimental Models to Study the Pathogenesis and Treatment of Mucormycosis"

_jof, 2024, doi:10.3390/jof10010085_

Round 1
Reviewer 1 Report
Comments and Suggestions for Authors
This is an excellent review by Ronen Ben-Ami about the experimental models to study the pathogenesis and treatment of mucormycosis, discusses the advantages and disadvantages of the different models as well as the unmet needs.
Author Response
I thank the reviewer for their kind remarks.
Reviewer 2 Report
Comments and Suggestions for Authors
The manuscript "Experimental Models to Study the Pathogenesis and Treatment of Mucormycosis" presents a very good narrative review, of model systems that have been established to study the pathogenesis and treatment of mucormicosis. The approach to the topic is clear, the presentation is good, and the references included are appropriate.
The manuscript presents a good review of the characteristics, applications, and limitations of various animal models of mucormycosis. My only suggestion is that it would be more illustrative for the reader if the author could include a table presenting the contributions to knowledge (of pathogenesis or treatment of mucormycosis) that have been achieved with each model used, since in the text it is not mentioned extensively, but is limited to explain how the model works.
The names of the microorganisms are in italics, please review the entire manuscript and correct
Author Response
I thank the reviewer for their kind remarks.
I added Table 2, which shows the contributions of various animal model systems to our understanding of mucormycosis.
All genus and species names have been italicized throughout the manuscript.
Reviewer 3 Report
Comments and Suggestions for Authors
This is a really interesting paper. It really sets out the pros and cons of each model. it sets it out in a very readable manner whilst maintaining the science. This will be a useful resource for years to come.
I think conclusions paragraph would be necessary for this paper.
Specifically:
Line 25 - should say that this occurred in India (at least predominantly).
Lines 27-28 - should say why mucormycosis is on the increase in these populations.
Author Response
I thank the reviewer for their kind remarks.
A conclusions paragraph was added to the manuscript.
Line 25: The sentence was modified with the occurrence of COVID-associated mucormycosis in India.
A sentence has been added (lines 29-31), speculating that increasing rates of mucormycosis may be attributable to a more immune-suppressed patient population, as well as a greater incidence of breakthrough fungal diseases in patients receiving mold-active antifungal prophylaxis.